# Peer review of "One-Size-Does-Not-Fit-All: The Case of Incremental Hemodialysis"

_kidneydial, doi:10.3390/kidneydial4010003_

Round 1
Reviewer 1 Report
Comments and Suggestions for Authors
Gaetano Casino and Basile present a very thoughtful and well-written review on the use of incremental dialyis in incident HD patients. The authors are clearly very much in support of using this approach to preserve RKF and thereby potentially improve quality of life and hard outcomes in this highly vulnerable patient population.
Except for these minor suggestions, I believe this manuscript should be accepted.
>the "2" in Beta-2 microglobulin should be subscript, rather than superscript (b2-microglobulin)
>page 5 line 226: Dialysis Center should not be capitalized
>the review is very favorable of incremental dialysis, which the authors underline well with arguments and data. However, the authors may wish to add one paragraph on criticism and potential pitfalls in incremental HD.
Reviewer 2 Report
Comments and Suggestions for Authors
The authors conducted a well-presented review on the benefits of incremental dialysis. The manuscript is wll written. I have only some minor comments which may improve the quality of the study:
1. Please add a Table summarizing the results of the studies analyzed in the the section Review of Literature
2. Please add a paragraph regarding the Limitations of incremental dialysis
3. In the Table 2 please add a footnote in order to help the reader understanding the content.
